

# Increased MCHC*RDW-SD interaction values: indicators of neurological impairment in lead-poisoned children

Qingji Ying[1], Mengsi Ye[1], Tingting Zhang[2], Zhaobo Xia[3] and Huale Chen[2]

[1] Department of Gastroenterology, The Second Affiliated Hospital and Yuying Children's Hospital of Wenzhou Medical University, Wenzhou, Zhejiang Province, China
[2] Department of Laboratory Medicine, The Second Affiliated Hospital and Yuying Children's Hospital of Wenzhou Medical University, Wenzhou, Zhejiang Province, China
[3] Department of Pediatric Surgery, The Second Affiliated Hospital and Yuying Children's Hospital of Wenzhou Medical University, Wenzhou, Zhejiang Province, China

## ABSTRACT

**Background**. The neurotoxic effects of lead in children can have long-lasting and profound impacts on the developing nervous system. This study aimed to identify a reliable and easily accessible biomarker to monitor neurological impairment in lead-poisoned children.

**Methods**. We analyzed hematological data from 356 lead-poisoned children, comparing them with age and gender-matched healthy controls. Multivariate logistic regression and receiver operating characteristic (ROC) analysis were employed to identify and evaluate potential biomarkers for neurological damage.

**Results**. Significant changes in erythrocyte parameters were observed in lead-poisoned children. Upon further analysis, increased mean corpuscular hemoglobin concentration (MCHC) and red cell distribution width-standard deviation (RDW-SD) interaction values were found to be significantly associated with neurological impairment. The MCHC*RDW-SD interaction model demonstrated an AUC of 0.76, indicating its effectiveness in reflecting neurological damage. Additionally, the MCHC*RDW-SD Interaction value showed weak or no correlation with other erythrocyte parameters, suggesting its independence as an indicator.

**Conclusion**. Our findings propose the increased MCHC*RDW-SD interaction value as a robust and independent biomarker for detecting neurological impairment in lead-poisoned children. This underscores the potential of utilizing specific erythrocyte parameters for screening the neurotoxic effects of lead exposure in pediatric populations.

Corresponding author
Huale Chen, hualechen@wmu.edu.cn

# INTRODUCTION

Historically, several small workshops in Wenzhou, including those specializing in paint, electroplating, and copper processing, have been significant sources of lead contamination. Lead exposure can have harmful effects, including impeding physical development, disrupting the hematopoietic system, altering immune responses, and affecting neuronal activity (*Nain & Kumar, 2020*; *Vorvolakos, Arseniou & Samakouri, 2016*;

*Zhang et al., 2016*). Lead poisoning can damage the hematopoietic system, affecting cerebral hemodynamics and mitochondrial function. This may disrupt brain autoregulation mechanisms, potentially leading to neurological damage (*Kallianpur et al., 2016*; *Wan et al., 2020*). Children are particularly vulnerable to lead due to their developmental stage and behaviors like hand-to-mouth actions. Long-term effects of lead exposure in children, such as cognitive and behavioral issues, are serious and often irreversible. Studies indicate that lead's neurotoxicity may increase the risk of criminal behavior in adulthood (*Ahmad et al., 2020*; *Wright et al., 2021*). Considering this, it is imperative to undertake early screening and intervention measures to protect children from neurological destruction caused by lead toxicity. Since 2010, our hospital has treated over 3,000 children with lead poisoning, many exhibiting neurological symptoms such as irritability, headaches, and memory impairments.

Analyzing erythrocyte parameters such as red cell distribution width (RDW), mean corpuscular volume (MCV), mean corpuscular hemoglobin (MCH), and MCHC has been effective in detecting various diseases, including cancer, injury, periodontitis, and venous thromboembolism (*Bruserud, Aarstad & Tvedt, 2020*; *Canet et al., 2015*; *França et al., 2019*; *Vayá & Suescun, 2013*). These parameters are also useful in identifying neurological impairments, although most studies have focused on adults (*Gong et al., 2019*; *Kallianpur et al., 2016*; *Lee et al., 2017*; *Oh et al., 2020*; *Wan et al., 2020*). Research on pediatric neurological conditions is more challenging due to children's limited cooperation and the variable reliability of traditional neurological tests (*Hobart, 2006*).

In this study, we discovered that a higher MCHC*RDW-SD interactions value indicates the presence of neurological damage in children poisoned by lead. We propose that MCHC*RDW-SD interactions value could be a promising marker for screening neurological damage in children affected by lead poisoning.

## MATERIALS & METHODS

### Patients

This study involved 356 children with lead poisoning (LP) and an equal number of age- and gender-matched healthy controls (HC), all of whom visited our hospital from January 1, 2015, to December 31, 2017. Participants for the LP group were selected based on two criteria: (i) a diagnosis of lead poisoning with a blood lead level of 100 μg/L or greater and (ii) being under 14 years of age. The LP group excluded patients with blood-related genetic diseases, such as hemoglobin genetic disorders, thalassemia, sickle cell anemia, or hemophilia, neurological symptoms caused by other heavy metals, such as mercury, cadmium, or chromium, and acute bacterial infection. Although initially, 360 lead-poisoned children were enrolled, three were excluded due to acute bacterial infection, and one due to hemophilia, thus resulting in a final LP group of 356 patients. We assessed neurological impairment through a retrospective review of medical records, employing the Ten Question Questionnaire for initial screening and subsequent detailed clinical and neurological examinations by pediatric specialists (*Abuga et al., 2022*; *Durkin, Hasan & Hasan, 1995*). We also recorded demographic and lead exposure information, including

age, gender, symptoms, and types of lead exposure, for the LP patients. The HC group comprised healthy children with blood lead levels <100 µg/L and no health issues detected in routine check-ups.

## MATERIALS AND METHODS

Peripheral venous blood was drawn from each child into two separate blood collection tubes for the measurement of hematology and blood lead levels. The Sysmex XE5000 hematology analyzer (Sysmex, Kobe, Japan) was used to automatically measure hematology parameters from whole blood. Additionally, the lead level in whole blood was semi-automatically determined by the Bohui BH2101s atomic absorption spectrometer (Bohui, Beijing, China). Briefly, the atomic absorption spectrometer was preheated for commissioning. The instrument's software automatically set the parameters to optimal conditions. A standard curve ($r \geq 0.9950$) was then constructed based on the absorbance values of the corresponding calibration solution. Finally, we added 40 µl of whole blood to 0.36 ml of diluent, mixed them well, and measured the blood lead concentration after half an hour at room temperature.

### Development of predictive models

In our study dataset, we included both hematology indicators with significant differences identified in this research and biochemical indicators with significant differences from another study previously published by our authors (*Ye et al., 2021*), both derived from the same cohort of participants, as continuous attributes. Additionally, types of lead exposure and neurological symptoms were included as categorical attributes. Neurological symptoms were used as the outcome variable, while all others served as independent variables. During data processing, we performed integer encoding on categorical variables for subsequent analysis. The dataset was randomly split into a training set (80%) and a validation set (20%). We selected three models for prediction: Random Forest, support vector machine (SVM), and generalized linear model (GLM), all trained using the full range of available features. The outcome variable for these models was the status of neurological damage in children with lead poisoning, classified based on clinical assessments and diagnostic results. Model performance was evaluated on the validation set by calculating the confusion matrix and receiver operating characteristic (ROC) curve, to select the best predictive model. We further utilized the recursive feature elimination (RFE) method in conjunction with the best predictive model to identify the most critical features for predicting neurological damage in lead-poisoned children. The final selection of features was based on their importance scores in the model. All analyses and model building were conducted in the R environment, using packages such as scikit-learn, randomForest, and e1071. We provided a detailed summary and coefficient analysis of the final logistic regression model, along with an ROC curve to demonstrate its performance. The best cut-off point for the indicator with optimal sensitivity and specificity was identified, followed by risk stratification and subgroup analyses.

## Statistical analysis

Data analysis was conducted using RStudio 4.3.0 (*RStudio Team, 2023*) and MedCalc 11.4.2 (https://www.medcalc.org/). Distribution normality was assessed using the Kolmogorov–Smirnov test, with results presented as mean ± SD for normally distributed data and median (inter-quartile range) for skewed data. Differences between the LP and HC groups were evaluated using independent Student t-tests for normally distributed variables and the Mann–Whitney U-test for non-parametrically distributed variables. Categorical variable differences were analyzed using the $\chi 2$ test. Statistically significant indicators were further analyzed with machine learning techniques. Spearman correlation analysis was employed to evaluate the independence of these indicators. Statistical significance was set at a *P*-value of less than 0.05.

The Medical Ethics Committee of the Second Affiliated Hospital of Wenzhou Medical University, Yuying Children's Hospital of Wenzhou Medical University granted Ethical approval to carry out the study within its facilities (Ethical Application Ref: 2022-K-106-02).

## RESULTS

This study thoroughly analyzed 356 children with lead poisoning and an equal number of healthy controls matched by age and gender. The median age for both groups was 4.0 years, with an inter-quartile range of 6.0 years. Males were the majority, comprising 72.11% (257) of each group. The mean blood lead level in the poisoned group was significantly higher at 186.47 µg/L, compared to 40.26 µg/L in the control group. Notably, 78.65% of the lead-exposed children exhibited neurological symptoms such as irritability, hyperactivity, headaches, and memory impairments. The sources of lead exposure were varied, including contaminated water (17.98%), traditional skin care products containing lead (23.88%), environmental pollution (21.07%), and parental occupational exposure (17.13%). In 17.98% of cases, exposure came from multiple sources, and 32.86% of the cases had unidentified sources of lead exposure.

A comparative analysis of hematology parameters indicated that children exposed to lead had significantly lower values of monocytes, basophil, hemoglobin, red blood cell count (RBC), hematocrit (HCT), MCH, MCHC, mean platelet volume (MPV), and platelet volume distribution width (PDW), and higher white blood cell count (WBC), neutrophils, eosinophil, red cell distribution width coefficient of variation (RDW-CV), RDW-SD, platelet, and plateletcrit (PCT) levels compared to healthy children (Table 1). These hematological differences, combined with biochemical indicators showing significant differences from another study previously published by our authors (*Ye et al., 2021*), which also utilized the same cohort of participants, were used to develop predictive models. The results from three different machine learning models indicated that the generalized linear model (GLM) algorithm was the most effective for classifying the presence of neurological impairment (Accuracy 0.84, Kappa 0.43, Table 2), incorporating variables such as MCHC, RDW-SD, RDW-CV, globulin, types of lead exposure, indirect bilirubin, and total bilirubin. Further assessment of the association between biomarkers and outcome

**Table 1  Hematologic parameters of the lead-poisoned children and the healthy controls.**

| Parameters | HC ($N = 356$) | LP ($N = 356$) | $P$ |
|---|---|---|---|
| WBC, $\times 10^9$/L | 7.97 ± 2.59 | 8.57 ± 3.01 | <0.05[*] |
| Neutrophils, $\times 10^9$/L | 3.43 ± 1.84 | 3.88 ± 1.67 | <0.05[*] |
| Lymphocytes, $\times 10^9$/L | 3.66 ± 1.56 | 4.06 ± 2.14 | 0.06 |
| Monocytes, $\times 10^9$/L | 0.44 ± 0.17 | 0.34 ± 0.19 | <0.05[*] |
| Eosinophil, $\times 10^9$/L | 0.25 ± 0.22 | 0.26 ± 0.32 | 0.20 |
| Basophil, $\times 10^9$/L | 0.04 ± 0.03 | 0.03 ± 0.02 | <0.05[*] |
| Hemoglobin, g/L | 131.17 ± 9.13 | 124.32 ± 10.72 | <0.05[*] |
| RBC, $\times 10^{12}$/L | 4.78 ± 0.36 | 4.65 ± 0.34 | <0.05[*] |
| HCT, % | 0.39 ± 0.03 | 0.37 ± 0.03 | <0.05[*] |
| MCV, fL | 81.29 ± 4.50 | 80.24 ± 6.40 | 0.08 |
| MCH, pg | 27.54 ± 1.74 | 26.85 ± 2.39 | <0.05[*] |
| MCHC, g/L | 338.68 ± 8.41 | 334.49 ± 10.61 | <0.05[*] |
| RDW-CV, % | 12.68 ± 0.86 | 13.54 ± 1.21 | <0.05[*] |
| RDW-SD, fL | 37.47 ± 2.29 | 43.52 ± 4.97 | <0.05[*] |
| Platelet, $\times 10^9$/L | 317.97 ± 68.47 | 338.63 ± 89.55 | <0.05[*] |
| PCT, % | 0.28 ± 0.06 | 0.29 ± 0.07 | 0.29 |
| MPV, fL | 9.04 ± 0.86 | 8.75 ± 1.20 | <0.05[*] |
| PDW, % | 15.53 ± 1.22 | 14.78 ± 1.95 | <0.05[*] |
| Blood lead, μg/L | 40.26 ± 15.60 | 186.47 ± 74.31 | <0.05[*] |

Notes.
Data are mean ± standard deviation.
[*]A statistically significant difference.
Abbreviations: WBC, white blood cell count; RBC, red blood cell count; HCT, hematocrit; MCV, mean corpuscular volume; MCH, mean corpuscular hemoglobin; MCHC, mean corpuscular hemoglobin concentration; RDW, red blood cell distribution width; CV, Coefficient of variation; SD, standard deviation; PCT, Plateletcrit; MPV, mean platelet volume; PDW, platelet volume distribution width.

**Table 2  Classification report of the machine learning algorithms for classifying neurological damage.**

| Algorithm | Accuracy | Kappa | Variables |
|---|---|---|---|
| Random forest | 0.83 | 0.41 | MCHC, RDW-SD, Types of lead exposure, RDW-CV, Globulin, Indirect bilirubin, Total protein |
| Support vector machine | 0.82 | 0.37 | MCHC, RDW-SD, Globulin, RDW-CV, Types of lead exposure, Total bilirubin, WBC |
| Generalized linear model | 0.84 | 0.43 | MCHC, RDW-SD, RDW-CV, Globulin, Types of lead exposure, Indirect bilirubin, Total bilirubin |

Notes.
Abbreviations: WBC, white blood cell count; MCHC, mean corpuscular hemoglobin; RDW, red blood cell distribution width; CV, Coefficient of variation; SD, standard deviation.

variables revealed that the coefficients for MCHC, RDW-SD, and types of lead exposure were statistically significant ($P$-value < 0.05), suggesting a strong association of these variables with neurological damage in children with lead poisoning Table 3.

To evaluate the effectiveness of various indicators in the model for detecting neurological damage in children with lead poisoning, receiver operating characteristic (ROC) analysis was performed. The results are graphically illustrated in Fig. 1. The values of each indicator

**Table 3**  Results of logistic regression analysis: assessing the association of biomarkers with outcome variables.

|  | Estimate | Std. Error | z value | Pr(>|z|) |
|---|---|---|---|---|
| (Intercept) | 24.49 | 6.82 | 3.59 | <0.05[*] |
| MCHC | −0.06 | 0.02 | −3.51 | <0.05[*] |
| RDW-SD | −0.16 | 0.05 | −2.93 | <0.05[*] |
| RDW-CV | 0.20 | 0.23 | 0.84 | 0.40 |
| Globulin | 0.09 | 0.05 | 1.84 | 0.07 |
| Types of lead exposure | −0.30 | 0.10 | −2.89 | <0.05[*] |
| Indirect bilirubin | 0.76 | 0.57 | 1.33 | 0.18 |
| Total bilirubin | −0.52 | 0.44 | −1.18 | 0.24 |

**Notes.**

Abbreviations: MCHC, mean corpuscular hemoglobin; RDW, red blood cell distribution width; CV, Coefficient of variation; SD, standard deviation.

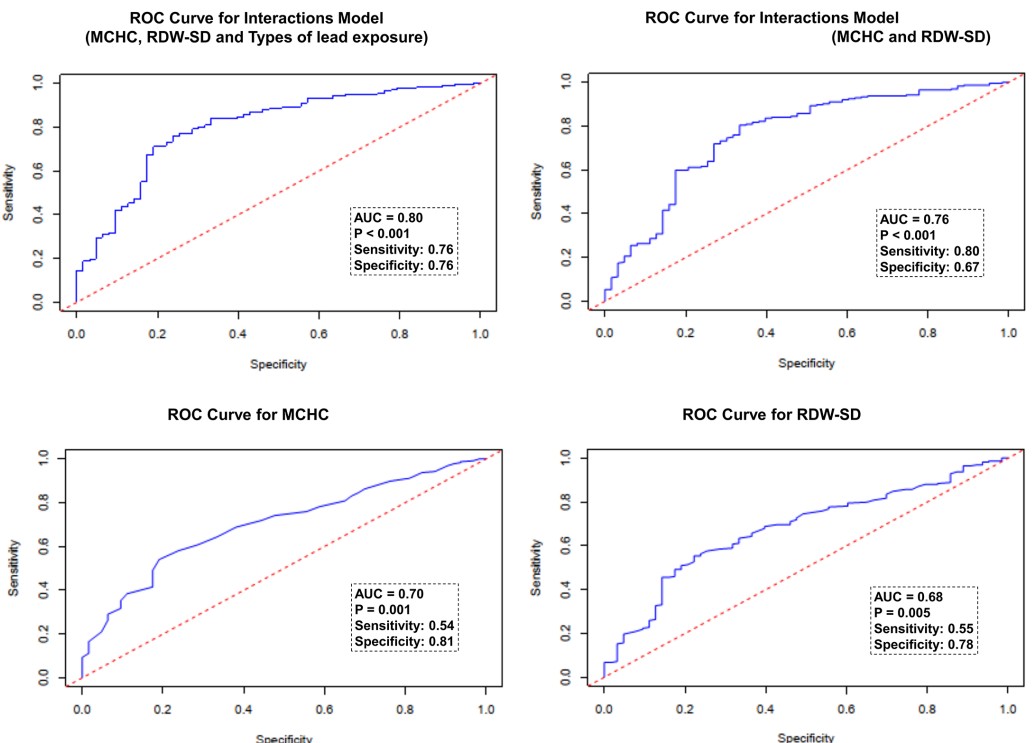

**Figure 1**  ROC analysis of the efficiency of indictors in reflecting the presence of neurological damage in lead-poisoned children.

along with their corresponding sensitivity and specificity are as follows: The highest performance was observed in the interactions model of MCHC, RDW-SD, and types of lead exposure, with an AUC of 0.80 (sensitivity 76%, specificity 76%). The interactions model of MCHC and RDW-SD showed a slightly lower AUC of 0.76 (sensitivity 80%,

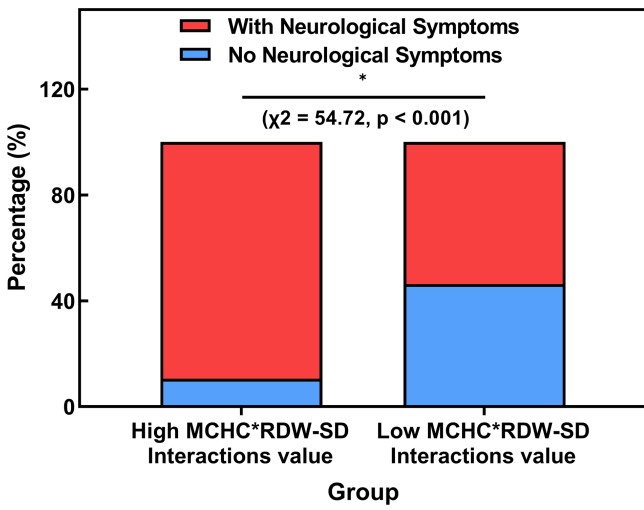

**Figure 2** Comparison of the rate of neurological impairment in lead-poisoned children groups divided by MCHC*RDW-SD Interactions cut-off value.

specificity 67%). For individual indicators, the AUC for MCHC was 0.70 (sensitivity 54%, specificity 81%), and for RDW-SD, it was 0.68 (sensitivity 55%, specificity 78%).

According to the ROC analysis, a criterion value of 1.046094 for the Interactions Model of MCHC and RDW-SD was identified to indicate neurological damage in children with lead poisoning. It was observed that 89.4% (220/246) of the lead-poisoned children with a higher MCHC*RDW-SD Interactions value (above 1.046094) exhibited neurological symptoms, significantly more than 53.7% (58/108) of those with a lower value ($\chi 2 = 54.72$, $p < 0.001$, Fig. 2).

Spearman correlation analysis indicated that the MCHC*RDW-SD Interactions value, apart from showing strong correlation with its parent indicators (MCHC and RDW-SD), exhibited no or weak correlation (less than 0.5) with other variables (Fig. 3).

Our analysis primarily focused on the interactions model of MCHC and RDW-SD. The rationale behind this focused approach, including the decision to exclude 'Types of lead exposure' from this stage of the analysis, is further elaborated in the discussion section.

## DISCUSSION

Accurately and early detecting neurological impairment in young children is challenging due to their limited compliance and the unpredictable nature of traditional neurological tests (*Hobart, 2006*). Our study finds that a significant 78.65% of children with lead poisoning exhibit neurological damage, thus highlighting the urgency for effective screening methods. Crucially, our study establishes a correlation between increased MCHC*RDW-SD interaction values and neurological damage in children with lead poisoning. The inherent advantages of MCHC and RDW-SD as objective, easily accessible, and reproducible hematological indicators lend them significant potential as reliable biomarkers for screening neurological damage in children affected by lead.

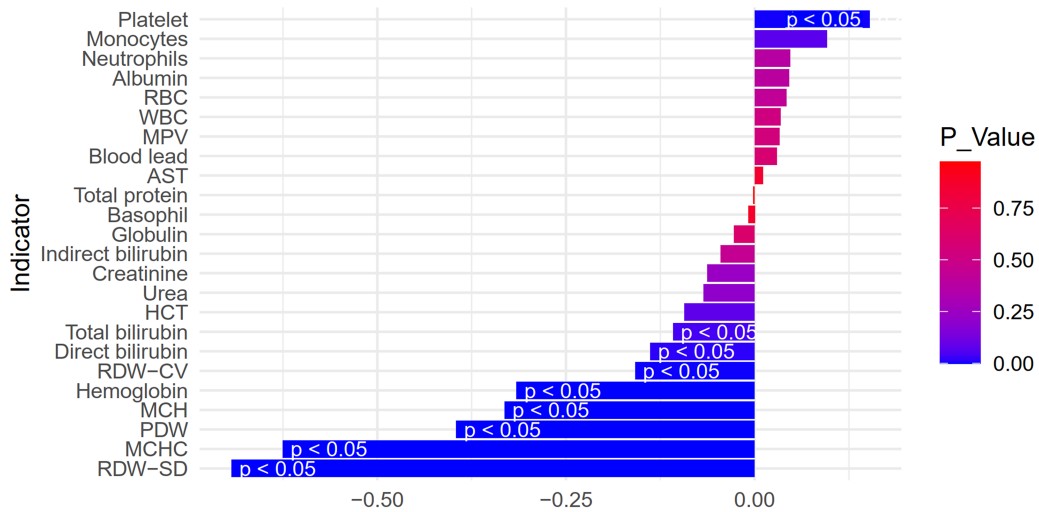

**Figure 3** **Spearman correlation of MCHC*RDW-SD interactions value and other parameters in lead-poisoned children.**

Lead toxicity is known to increase the release of premature erythrocytes into circulation, which have a diminished capacity for oxygen transport (*Kiefer & Snyder, 2000*; *Nishiyama et al., 2016*). Adequate oxygenation is critical for cerebral metabolism and the repair of neurological damage (*Wang et al., 2020*). Based on this understanding, we hypothesized that erythrocyte abnormalities might be indicative of neurological damage in children with lead poisoning. Indeed, our results identified a correlation between neurological impairment symptoms in these children and abnormal erythrocyte indicators, notably elevated MCHC*RDW-SD interaction values. This correlation suggests that these indicators, reflecting individual erythrocytes' hemoglobin concentration, could serve as a more precise measure of cerebral oxygenation, especially considering the oxygen transport mechanism through the blood–brain barrier (*Kornhuber et al., 1987*; *Lee et al., 2017*).

Numerous studies have established a correlation between abnormal hematological indices and neurological damage in adult patients. For example, elevated mean corpuscular hemoglobin concentration (MCHC) levels have been significantly associated with various neurological and cognitive impairments in adults, as demonstrated in research studies (*Gong et al., 2019*; *Kallianpur et al., 2016*; *Lee et al., 2017*; *Oh et al., 2020*). Furthermore, abnormalities in red cell distribution width (RDW) have been linked to neurological conditions in adults (*Raymond et al., 2009*). However, it's crucial to recognize that children's hematological profiles are markedly different from those of adults (*Chwalba et al., 2018*). *Raymond et al. (2009)* observed that while RDW levels are significantly lower in lead-exposed adults, these levels are noticeably higher in 3- and 4-year-old children exposed to lead, highlighting the differential impact of lead across age groups (*Raymond et al., 2009*). Given that our study focused on pediatric participants with lead-induced neurological damage, it raises the question of whether MCHC and RDW are as effective in detecting neurological damage in children with lead poisoning as they are in adults.

In our model, the combination of MCHC, RDW-SD, and types of lead exposure was found to be a strong predictor of neurological impairment in lead-poisoned children, with an area under the curve (AUC) of 0.80. However, with the aim of identifying objective hematological indicators, we excluded the variable of lead exposure types and developed a model based solely on the interaction of MCHC and RDW-SD. This model also demonstrated considerable predictive ability for neurological impairment (AUC 0.76). With the cut-off value for the MCHC*RDW-SD interaction set at 1.046094, we observed a significantly higher proportion of neurological damage in children within the high MCHC*RDW-SD interaction group (89.4%) compared to the low interaction group. Furthermore, apart from MCHC and RDW-SD, no other erythrocyte parameters showed more than a weak correlation (less than 0.5) with the MCHC*RDW-SD interaction value, indicating its relative independence as an indicator. These findings suggest that an increased MCHC*RDW-SD interaction value is a reliable marker for the presence of neurological damage in children with lead poisoning.

As the premier children's hospital in southern Zhejiang, our institution has become the primary destination for consultations and treatment of lead-poisoned children, as well as for routine physical examinations. This position has enabled us to conduct a large case-control study with a substantial sample size. However, there are some limitations to consider. First, our study does not represent a long-term clinical validation; while adjustments and analyses were made for age and gender—factors known to influence hematologic parameters in children (*Bohn et al., 2020*; *Higgins et al., 2020*; *Tahmasebi et al., 2020*)—other potential confounders such as medication and nutritional status were not considered. Second, the out-migration of workshops since 2018 has led to a decrease in the number of new cases of lead-poisoned children, which may affect the contemporaneity of our research data. Despite these limitations, our findings offer valuable insights for the screening of neurological damage in lead-poisoned children, which can be applied in other developing industrial regions.

## CONCLUSIONS

In conclusion, our study identified a significant association between increased MCHC*RDW-SD Interaction values and the commonly observed neurological impairment in children with lead poisoning. The straightforward and reliable measurement of MCHC and RDW-SD shows potential for use in the early detection of neurological damage in this vulnerable group. However, further research is necessary to fully establish the potential of these indicators in clinical screening protocols.

### Funding

This study was supported by the Wenzhou Municipal Science and Technology Bureau Foundation of China under grant No. Y2020246, the Zhejiang Provincial Natural Science Foundation of China (grant number LGF19H190002) and the Zhejiang Provincial

Medical and Health Science and Technology Science Foundation of China (grant number 2019RC211). The funders had no role in study design, data collection and analysis, decision to publish, or preparation of the manuscript.

## Grant Disclosures

The following grant information was disclosed by the authors:
Wenzhou Municipal Science and Technology Bureau Foundation of China: Y2020246.
Zhejiang Provincial Natural Science Foundation of China: LGF19H190002.
Zhejiang Provincial Medical and Health Science and Technology Science Foundation of China: 2019RC211.

## Competing Interests

The authors declare there are no competing interests.

## Author Contributions

- Qingji Ying performed the experiments, analyzed the data, prepared figures and/or tables, authored or reviewed drafts of the article, and approved the final draft.
- Mengsi Ye performed the experiments, analyzed the data, prepared figures and/or tables, authored or reviewed drafts of the article, and approved the final draft.
- Tingting Zhang performed the experiments, analyzed the data, prepared figures and/or tables, authored or reviewed drafts of the article, and approved the final draft.
- Zhaobo Xia performed the experiments, analyzed the data, prepared figures and/or tables, authored or reviewed drafts of the article, and approved the final draft.
- Huale Chen conceived and designed the experiments, analyzed the data, authored or reviewed drafts of the article, and approved the final draft.

## Human Ethics

The following information was supplied relating to ethical approvals (i.e., approving body and any reference numbers):

Medical Ethics Committee of the Second Affiliated Hospital of Wenzhou Medical University, Yuying Children's Hospital of Wenzhou Medical University granted Ethical approval to carry out the study within its facilities (Ethical Application Ref: 2022-K-106-02)

## Data Availability

The data is available in the Supplementary Files and at figshare: Ying, Qingji (2023). clinical data. figshare. Dataset. https://doi.org/10.6084/m9.figshare.24465856.v1.

## Supplemental Information

Supplemental information for this article can be found online at http://dx.doi.org/10.7717/peerj.17017#supplemental-information.

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
