# Peer review of "Increased MCHC*RDW-SD interaction values: indicators of neurological impairment in lead-poisoned children"

_PeerJ, doi:10.7717/peerj.17017_

## Round 0.1 · original submission · Major Revisions

Please, address the reviewers comments comprehensively. I have doubts about your methodology: why the focus in MCHC without previously explain why all other markers may have been disregarded? Why didn't you do a more comprehensive analysis, that included symptoms and/or other haematological / biochemical markers? It may be a question of language, clearly stating your general and specific objectives but still... very important, when this is a topic already well debated in literature. Proper data presentation, its completeness and transparency also need to be assured!

Reviewer 1 ·

Basic reporting

In general, figures and tables have been presented to support this manuscript. But there are several suggestions for the author to make the manuscript even better.

Line 46-48: The deleterious effects of lead can be devastating, causing hindrance in physical development, disrupting the hematopoietic system, altering immune responses, and affecting neuronal activity, amongst others.

Please explain in more detail about the relationship between Lead (Pb) – disrupting the hematopoietic system – potential indicators of neurological impairment.
Because in the Introduction section, there is not enough information about the relationship between these variables

Line 64-67: In this study, we discovered that a lower mean corpuscular hemoglobin concentration (MCHC) indicates the presence of neurological damage in children poisoned by lead. We propose that MCHC could be a promising marker for screening neurological damage in children affected by lead poisoning

Please explain why only MCHC is considered important as an indicator of neuronal disorders.
What about other hematological parameters (Hb, Erythrocytes, MCV and MCH)?

Experimental design

No comment

Validity of the findings

No comment

Additional comments

Line 214-220: In conclusion, we found that lower MCHC is common in lead-poisoned children, and a greater decrease in MCHC is significantly associated with neurological impairment. Given the ease of obtaining MCHC and its stability, as well as the urgent need to detect neurological damage in lead-poisoned children, we suggest that screening measures for neurological impairment should focus on low MCHC levels. However, further prospective interventional studies are needed to confirm the effectiveness of MCHC in indicating the presence of neurological damage in lead-poisoned children.

Is the MCHC value alone so that we can be said that a decrease in the MCHC value can be used as an indicator of neuronal disorders?
It would be nice if there were other accompanying data besides haematological parameters.

Annotated reviews are not available for download in order to protect the identity of reviewers who chose to remain anonymous.

Reviewer 2 ·

Basic reporting

no comment

Experimental design

no comment

Validity of the findings

1.The multivaraite analysis did not included symptoms, and types of lead exposure but only inlcued age and gender.
2. Figure 2 should indicat significance level
3.the diagnostic abiliyt is very low. The AUC was only 0.695. See if this MCHC can be used for diagnosing lead-posoned. I belive the AUC will be good one.

Additional comments

no comment

Reviewer 3 ·

Basic reporting

.

Experimental design

.

Validity of the findings

.

Additional comments

This retrospective hospital record review of 356 lead exposed children (BLL >100ug/L) identified hematological parameters that were associated with neurological symptoms, specifically, mean corpuscular hemoglobin concentration (MCHC). I think case-control studies of children and lead exposure with meaningful clinical outcomes can make important contributions to the literature on lead. However, I found this study and write-up difficult to interpret. More information and clarity is needed to determine whether readers in the field will find the results informative.

Major comments

First, the most critical issue that needs to be addressed is whether these new hematological parameters add anything of value over and above blood lead level (BLL). BLL has been robustly associated with a host of meaningful neurocognitive outcomes, such as IQ. Does MCHC add anything of value over and above BLL? Why not, in other words, just use BLL? If this paper is to be useful, this issue must be addressed. I would be happy to publish a null finding if the answer is “no hematological information is more useful than BLL.”

Second, the language throughout the manuscript needs a careful edit. Primarily, the language is often florid, non-scientific, and imprecise. For example, words like “insidious,” “devastating,” “diminutive,” and “bygone” are not appropriate for a scientific empirical article. The high frequency of literary words in the manuscript, which are never used in scientific writing and are very rarely used in everyday speech, make me suspect that the manuscript was written using generative AI or an AI translation service.

Abstract and Introduction

Abstract and Introduction lacks important details and precision. What counts as “lead-poisoned”? What was the source of lead, if known? What makes you say the control group was “healthy”? What were the blood lead levels (BLLs) in the two groups? What is the measure of “neurological symptoms”?

I am not convinced that “Research involving children’s neurological conditions is more complex than in adults, as young children have limited compliance, traditional neurological tests are challenging to fully assess due to their unpredictable repeatability.“ Which tests are you referring to, exactly? And for what neurological conditions? It is not difficult to test children’s cognitive ability or cognitive development. This is routinely done, which extremely good test-retest reliability.

Methods

The methods needs more details. Where were the “healthy” children recruited from? What was their prevalence of neurological symptoms?

What is the Ten Questions Questionnaire? What does it mean that a clinical exam was used “to confirm the diagnosis”? Were diagnoses made via a 10-item questionnaire? What diagnoses does this produce? What qualified children to be “neurologically damaged” versus not?

How were the sources of lead exposure identified?

The statistical analysis section needs more detail. How were the exposure and outcome variables treated in the models? The logistic regression suggests that the hematological measures were treated as dichotomous predictor variables. Is this true? What cut-points were used? Are these clinically meaningful? Why not use these measures as continuous?The same questions exist for the neurological outcome measure.

Please add BLLs to Table 1

---

## Round 0.2 · accepted · Accept

Dear authors, I am happy to let you know I am accepting your manuscript at this stage. I'd still like to emphasize the importance of careful proofreading, in the production stage. Thank you for your submission to PeerJ.

Reviewer 2 ·

Basic reporting

No comment

Experimental design

No comment

Validity of the findings

No comment

Additional comments

No comment

Reviewer 3 ·

Basic reporting

The writing is much improved. It is not perfect but it is in the realm of acceptable. Overall the language is clear, with sufficient rationale and literature review, and clear tables.

Experimental design

I believe the analyses have been conducted appropriately and the design reasonable given the limitations inherent in lead exposure research,

Validity of the findings

I have no new validity concerns for this revision.

Additional comments

This revision has addressed a number of my most significant concerns. I feel the new manuscript is easier to read, interpret, and understand.

I suggest the author substitute “neurological injury” for “neurological destruction” in the Introduction.

I suggest the authors include in the manuscript itself one or two sentences of the additional details on the TQQ that were provided to this reviewer.

I am not sure most readers will be familiar with the acronym “MCHC*RDW-SD” and wonder if the title can be amended to something that will be clear to a wide audience.